# Exploring the Drought Tolerant Quantitative Trait Loci in Spring Wheat

**DOI:** 10.3390/plants13060898

**Published:** 2024-03-21

**Authors:** Zhong Wang, Xiangjun Lai, Chunsheng Wang, Hongmei Yang, Zihui Liu, Zheru Fan, Jianfeng Li, Hongzhi Zhang, Manshuang Liu, Yueqiang Zhang

**Affiliations:** 1Institute of Nuclear and Biological Technologies, Xinjiang Academy of Agricultural Sciences, Urumqi 830091, China; zhongwang@xaas.ac.cn (Z.W.); wangchunsheng@xaas.ac.cn (C.W.); zherufan@xaas.ac.cn (Z.F.); hssljf@xaas.ac.cn (J.L.); dreamzhz@xaas.ac.cn (H.Z.); 2Key Laboratory of Crop Ecophysiology and Farming System in Desert Oasis Region, Ministry of Agriculture, Institute of Nuclear and Biological Technologies, Urumqi 830091, China; 3Xinjiang Key Laboratory of Crop Biotechnology, Institute of Nuclear and Biological Technologies, Urumqi 830091, China; 4Xinjiang Crop Chemical Control Engineering Technology Research Center, Institute of Nuclear and Biological Technologies, Urumqi 830091, China; 5State Key Laboratory of Crop Stress Biology for Arid Areas, College of Agronomy, Northwest A&F University, Yangling 712100, China; lxj2820@nwafu.edu.cn; 6Institute of Applied Microbiology, Xinjiang Academy of Agricultural Sciences, Urumqi 830091, China; yanghm@xaas.ac.cn; 7Xinjiang Laboratory of Special Environmental Microbiology, Institute of Applied Microbiology, Urumqi 830091, China; 8Department of Biochemistry, Baoding University, Baoding 071000, China; lzh@nwafu.edu.cn

**Keywords:** quantitative trait locus, agronomic traits, drought tolerance, wheat, yield stability

## Abstract

Drought-induced stress poses a significant challenge to wheat throughout its growth, underscoring the importance of identifying drought-stable quantitative trait loci (QTLs) for enhancing grain yield. Here, we evaluated 18 yield-related agronomic and physiological traits, along with their drought tolerance indices, in a recombinant inbred line population derived from the XC7 × XC21 cross. These evaluations were conducted under both non-stress and drought-stress conditions. Drought stress significantly reduced grain weight per spike and grain yield per plot. Genotyping the recombinant inbred line population using the wheat 90K single nucleotide polymorphism array resulted in the identification of 131 QTLs associated with the 18 traits. Drought stress also exerted negative impacts on grain formation and filling, directly leading to reductions in grain weight per spike and grain yield per plot. Among the identified QTLs, 43 were specifically associated with drought tolerance across the 18 traits, with 6 showing direct linkages to drought tolerance in wheat. These results provide valuable insights into the genetic mechanisms governing wheat growth and development, as well as the traits contributing to the drought tolerance index. Moreover, they serve as a theoretical foundation for the development of new wheat cultivars having exceptional drought tolerance and high yield potentials under both drought-prone and drought-free conditions.

## 1. Introduction

Wheat (*Triticum aestivum* L.) is an important staple crop, because it feeds more than 35% of the world’s population [1]. To meet the associated rising food demand, it is estimated that crop production needs to grow by at least 2.4% annually [2]. Thus, improving the grain yield potential is still a major task in wheat breeding, and it can be realized by improvements in the three main grain yield components, i.e., spike number per unit area, grain number per spike (GNPS) and thousand-grain weight (TGW) [3]. Moreover, other agronomic traits also play important roles in the determination of wheat yield, such as plant height (PH), spike length (SL), grain weight per spike (GWPS), grain length (GL) and grain width (GW) [4].

Potential yield is closely associated with plant photosynthesis [5]. Genetic improvement of grain yield components and physiological traits can certainly increase total yield. Quantitative trait loci (QTLs) mapping is a key approach for understanding the genetic architecture of yield components and physiological traits in wheat [6]. Previously, QTL mapping using various segregating populations was conducted for plant height, spike length, spike number per unit area, grain number per spike and thousand-grain weight [7,8,9,10,11]. However, QTLs were defined by relatively large genetic distances due to the limited numbers of markers. In addition, QTLs for physiological traits were rarely reported, except a few association studies for Soil and Plant Analyzer Development (SPAD) values of chlorophyll content, normalized differential vegetation index (NDVI) and canopy temperature (CT) in spring wheat [12,13,14].

However, climate stress and depleting fresh water for agricultural irrigation have severely affected wheat production, and drought stress is a major threat to wheat yield [15]. Wheat is particularly susceptible to drought-induced stress throughout the growth period; therefore, mining drought-stable QTLs is vital for increasing wheat yield. The drought susceptibility index (DSI) is used to measure yield stability in wheat genotypes that captures the changes in both drought stress and non-stress environments [16], whereas DSI values of <1 indicate tolerance to drought stress [17]. Genotypes with high yield stability index [18] and relative drought index [19] values are generally regarded as stable under stress and non-stress conditions. Similarly, the stress tolerance index [20] and geometric mean productivity [21] are useful indices for the identification of stable genotypes, which produce high yields under drought stress and higher or optimum yields under non-stress conditions.

QTLs have been detected for grain yield-related drought-tolerance indices traits in wheat [12,22,23]. However, research on the identification of QTLs associated with drought-tolerance indices of traits other than grain yield is scarce. For instance, the QTLs associated with drought indices calculated from total yield (TY), thousand-grain weight and grain number in durum wheat have been identified [24]. Similarly, the QTL-rich regions associated with drought indices derived from grain yield, thousand-grain weight and grain number per spike in bread wheat were identified [25].

In this research, a consensus genetic map was constructed using one bi-parental population of RILs in wheat to screen the genotypic variations in the agronomic index and drought tolerance index (DI) under different water-stress conditions. The main goal was to identify growth and development, as well as drought-tolerant, relevant QTLs in wheat. The findings will improve the understanding of the genetic mechanisms of wheat growth and development, as well as drought-tolerance index, traits and will provide new genetic loci for breeding new high-yielding and stress-tolerance wheat varieties.

## 2. Results

### 2.1. Phenotypic Data Analysis

In total, 18 yield-related agronomic traits and physiological traits of the RILs and parents were investigated in the field under non-stress and drought-stress conditions, and they displayed a continuous distribution (Appendix A), indicating that these traits were controlled by multiple genes in this population. The differences between non-stress and drought-stress conditions reached a significant level (*p* < 0.01) for all the traits, except the heading date (HD) (Appendix A, Table 1). Therefore, the wheat in the drought-stress trials was subjected to drought stress for almost the entire growth period that affected each trait examined. In particular, grain weight per spike (GWPS) and total yield (TY) exhibited the most significant decreases under drought-stress conditions (−16.8% and −14.1%, respectively, *p* < 0.01), but the number of basal sterile spikelet (BSS) was most significantly increased (26.9%, *p* < 0.01) (Table 1), indicating that the key factors directly limiting yield were particularly sensitive to drought stress.

Pearson’s coefficients of correlation for all the traits were analyzed under different conditions. Compared with the non-stress condition, total yield was significantly negatively correlated with grain number per spike, early filling stage canopy temperature (EF-CT) and middle filling stage canopy temperature (MF-CT) under drought-stress conditions, but there was no significant correlation between total yield and plant height, heading date chlorophyll content (HD-SPAD), EF-SPAD, elongation stage normalized differential vegetation index (ES-NDVI), HD-NDVI and MF-NDVI (Figure 1). Drought stress may change the relationships between yield and other traits in wheat.

### 2.2. QTL Analysis of Agronomic Traits

A total of 1027 SNP markers established on all the chromosomes (except 4D) were used for linkage map construction, producing a total map length of 2188.35 cM. The A, B and D sub-genomes harbored different numbers of SNPs, the 462 (0.63 per cM), 471 (0.51 per cM) and 114 SNP (0.22 per cM), respectively. The genetic map was used to identify significant associations between SNP markers and agronomic traits (Table 2).

QTL mapping was performed for all 18 yield-related agronomic traits and physiological traits under non-stress and drought-stress conditions. In total, 131 QTLs for the 18 traits were identified across the non-stress and drought-stress conditions, and they explained 0.25–45.11% of the phenotypic variances (Appendix A). Among them, 28 repetitive QTLs in more than two environments were defined as reliable loci (Table 3).

### 2.3. Drought Tolerance Evaluation

The drought tolerance index (DI) was used to assess the drought tolerance of the RIL population based on non-stress and drought-stress conditions (Figure 2, Appendix A). For yield-related agronomic traits, the DIs of grain weight per spike (GWPS) (0.85) and total yield (TY) (0.87) were the lowest, confirming that they were the most vulnerable traits to drought stress, resulting in severe adverse effects on wheat yield (Figure 2). The DI of number of basal sterile spikelet (BSS) (1.36) was higher than those of other yield-related traits, indicating that BSS was greatly affected by drought stress and phenotypic varied greatlyunder different conditions (e.g., no-stress/drought-stress and different years). For physiological traits, the DIs of NDVI in filling stage were significantly difference from other physiological traits, especially in the MF (only 0.73), indicating that the drought tolerance of NDVI in the filling stage was the weakest. Thus, drought stress appears to negatively affect grain development, especially grain formation and grain filling, directly leading to decreases in grain weight per spike (GWPS) and total yield (TY).

### 2.4. QTL Analysis of the DI

The annual values of the DI of each trait were calculated by combining the phenotypic data from 3 years, and these were then used for QTL detection. There were 20 QTLs related to drought tolerance among yield-related agronomy traits and 23 QTLs related to drought tolerance among yield-related physiological traits. They explained 5.62–53.65% of the phenotypic variances (Figure 3 and Appendix A).

Six QTLs were identified as being directly related to drought tolerance in wheat (Figure 4). To verify the drought tolerance roles of these vital QTLs, we performed a polymorphic characterization of the associated SNP markers in the RIL population and examined whether the differences in phenotypic values grouped by polymorphism exceeded a critical threshold (*p* < 0.05). We identified superior alleles as those associated with high DIs of thousand-grain weight, grain weight per spike and total yield, and inferior alleles as those having the opposite effects. Accessions with superior alleles for DIs of thousand-grain weight, grain weight per spike and total yield at these QTLs showed significantly higher than those with inferior alleles (Figure 5), indicating that the superior alleles of these QTLs could maintain the stability of thousand-grain weight, grain weight per spike and total yield under drought-stress conditions. In conclusion, the superior alleles of these loci could be integrated into wheat cultivars by marker-assisted selection for breeding to improve the drought tolerance and increase the stability of yield-associated traits under drought-stress conditions.

## 3. Discussion

Owing to climate change, the occurrence of drought stress during crop growth is becoming a major obstacle to yield improvement [27,28]. Thus, it is crucial to assess the adaptability of wheat to drought stress under future climate conditions [29,30]. This study found that drought stress significantly reduced the yield-related agronomic and physiological traits of wheat, especially total yield and yield components, such as thousand-grain weight and grain weight per spike, compared with under non-stress conditions (Appendix A). The greatest impact of drought on total yield may be partly due to the cumulative effects it exerts on the yield-related traits, as well as on the flowering and grain filling stages [31,32]. For instance, drought stress caused a significant reduction in tiller number, ovary pollination and spike size [33,34]. Thus, drought stress also changed the relationship between wheat yield and other traits, especially the correlation between yield and physiological traits [35]. Consequently, improving the tolerance of yield component-related traits to drought stress is the basis for maintaining wheat yield levels under extreme weather conditions.

Genetic mapping is an important method to analyze the genetic mechanisms of wheat traits. Here, we used 1047 polymorphic SNP markers from a 90K SNP assay [36] and constructed a high-density genetic map for an RIL population. The average density of the map was 2.09 cM/marker, representing a considerable improvement over previously reported maps [37,38]. In total, 131 QTLs were identified for 18 yield-related agronomic and physiological traits. There were 67 QTLs (51.1%) located on the A genome, 44 QTLs (33.6%) located on the B genome, and 20 QTLs (15.3%) located on the D genome, which was basically consistent with the distribution of markers on the genetic marker map. Many QTLs are pleiotropic [39,40], which was corroborated in the present study, with 27 QTLs being associated with more than two traits. These findings improve our understanding of the genetic mechanisms of wheat growth and development, and they provide a basis for further genetic analysis of wheat yield and physiological traits.

From the experimental procedures under both non-stress and drought-stress conditions, the traits used to calculate the drought tolerance index (DI) were significantly different among wheat materials and treatments. This illustrates the broad genetic diversity present in the materials used to promote drought tolerance in general [40]. This relationship provides the basis for utilizing DIs as means to explain phenotypic variation. In this study, the DIs of grain weight per spike and total yield, the most susceptible yield traits to drought stress, were only 0.85 and 0.87, respectively. The DI-NDVI was only 0.73 at middle filling stage, indicating that drought stress at this stage had an adverse effect on grain formation and grain filling, which directly contributed to drought-induced decreases in grain weight per spike and total yield.

Increasing the tolerance of wheat to drought stress through genetic improvement strategies is of great significance to ensure food security [41]. The drought tolerance index (DI) can be derived from total yield and strongly positively correlated traits, and it is used as a measure to select the best genotype [42,43,44]. A total of 43 QTLs associated with trait-related drought tolerance were identified using the drought tolerance index (DI) of each trait. Among them, six were directly associated with drought tolerance in wheat. Despite the identification of several QTLs that were associated with drought indices in our study, further validations and investigations are needed to understand the molecular functions of the associated genes in wheat drought stress-response mechanisms. Major QTLs with favorable SNP alleles identified in this study could be used to develop markers, such as cleaved amplified polymorphic sequences and competitive allele-specific PCR-based markers, to facilitate future marker-assisted breeding in wheat.

## 4. Materials and Methods

### 4.1. Plant Materials and Field Trials

A total of 186 F_8_ RILs derived by single-seed descent (SSD) from a cross between XinChun7 (XC7) and XinChun21 (XC21) were used for phenotyping and QTL mapping. XC7 is characterized by high yield, excellent bread quality, good lodging resistance and drought-stress tolerance. XC21 is characterized by high yield as well, while being sensitive to drought stress.

The parents and 186 RILs were grown at Xinjiang Wheat Breeder Base in Chang Ji District (43°96′ N, 87°01′ E, altitude 717.2 m) from 2014 to 2016. The field trials were carried out in randomized complete blocks with three replications and two conditions, drought-stress and non-stress conditions. The irrigation method for both conditions was drip irrigation, which was independently controlled for the different treatments, and the areas under the different treatments were separated by 4 m isolation zone. Each replicate experimental plot was 2 m in length, with four rows, and an inter-row spacing of 0.2 m, with 40 grains per row. Field management was consistent with local practices for wheat production. No-stress condition plants were watered eight times during the growing season with irrigation intervals of 10 days. Drought-stress condition plants were irrigated three times at the jointing, heading and early grain filling stages. Non-stress and drought-stress condition plants had 480 mm and 180 mm irrigation water applied, respectively. During the 2014, 2015, and 2016 growing seasons, both treatments received 65, 74.5, and 142 mm of rainfall, respectively.

### 4.2. Phenotypic Evaluation

The traits were measured at physiological maturity. Three individual plants were selected from each row and used to investigate plant height, grain number per spike and number of basal sterile spikelets in accordance with established protocols [45,46]. The heading date was assessed as the interval between the date of seeding emergence and the date at which 50% of spikes per row emerged from the flag leaf [47]. The grain fullness was divided into nine grades, from very shrunken (1) to very full (9). In addition, thousand-grain weight and grain weight per spike were recorded using the rapid SC-G grain appearance quality image analysis system [45]. Finally, all the plants in the plot were harvested manually, and after threshing, the weight was calculated as the total yield (t/ha) when the water content was 13%.

The canopy temperature was measured at 13:00–15:00 using a hand-held thermal infrared instrument (FLIR Integrated Imaging Solutions Inc., Richmond, BC, Canada) during the heading stage, early filling stage (9 days after flowering) and middle filling stage (18 days after flowering), with a field of view of 25° × 20° and a resolution of 320 × 240 [48]. The SPAD was measured in 10 flag leaves per plot at the heading stage, early filling stage and middle filling stage using a SPAD-502 Minolta chlorophyll meter (Spectrum Technologies, Plainfield, IL, USA). Three points were taken per leaf, and the three readings were averaged [48]. The NDVI was measured during the elongation stage, heading stage, early filling stage and middle filling stage using a Green Seeker 505 (Hand Held Optical Sensor Unit, Model 505; NTech Industries Inc., Ukiah, CA, USA) [49].

### 4.3. Statistical Analyses

The phenotypic data analyses were conducted using Genstat v.17.1 (VSN International Ltd., Hemel Hempstead, UK) and SPSS Statistics v.20.0 (IBM Corporation, Armonk, NY, USA). PROCGLM was used for the analysis of variance, in which genotypes were considered as fixed effects, and environments and replicates nested in environments were considered as random effects. Pearson’s correlation analysis and independent-samples *t*-tests were performed using the IBM SPSS Statistics version 20.0 software (IBM Corporation, Armonk, NY, USA).

The drought-resistance coefficient (DC) and DI were used to assess the drought tolerance of wheat [16,50]. The formulas are as follows:*DC* = *P*_*d*_/*P*_*i*_
*DI* = *DC* × *P*_*d*_/*P*_*d*_,
where *P_d_* represents the phenotype under drought conditions; *P_i_* represents the phenotype under irrigated conditions; and *P_d_* represents the average phenotype under drought conditions.

### 4.4. SNP Genotyping

Total genomic DNA was isolated from the lyophilized young leaves of each genotype using the DNeasy Plant Mini Kit (Qiagen, Valencia, CA, USA; cat. no. 69106). The DNA quality was assessed using 0.8% agarose gel electrophoresis, and the DNA concentration was measured using a NanoDrop™ ND-2000 spectrophotometer (Thermo Fisher Scientific Inc., Waltham, MA, USA). The RIL, XC7 and XC21 accessions were genotyped using the Illumina 90K iSelect wheat SNP assay [36] in the Small Grains Genotyping Lab at the USDA-ARS, Fargo, ND, USA. The Illumina iSelect 90K assay produced data for 81,587 SNPs. The analyses of SNP genotyping, clustering of the SNP alleles and calling of the genotypes were performed using Genome Studio v2011.1 (https://www.illumina.com, accessed on 5 June 2023). The minimum number of points used in a cluster was 10 [36]. Monomorphic SNPs, and SNPs having more than 20% missing genotypic data and 10% heterozygosity, were excluded. The polymorphic SNPs selected after filtering based on the above-mentioned criteria were screened to determine their positions on the chromosomes based on the wheat consensus genetic map [36].

### 4.5. Genetic Map Construction and QTL Mapping

For genetic map construction, the monomorphic markers between parents and the markers with a high missing value (i.e., more than 20.0%) or minor allele frequency less than 0.3 were removed, and the remaining 11,213 high-quality polymorphic markers were used for subsequent analyses. The BIN function in IciMapping v4.2 (http://www.isbreeding.net/, accessed on 5 November 2023) [51] was used to remove redundant markers. A linkage analysis of the 4784 non-redundant markers was performed with the JoinMap v4.0 Software using the regression mapping algorithm. The linkage maps were drawn using the IciMapping v4.2 Software.

A QTL analysis was carried out using the inclusive composite interval mapping function of IciMapping v4.2. The mapping parameters were set as step = 0.1 cM and PIN = 0.001, and the logarithm of odds threshold was calculated with 1000 permutations at *p* < 0.05. QTLs that explained more than 5% of the phenotypic variance were considered to be major loci, and those detected in more than two environments were regarded as stable.

## 5. Conclusions

We evaluated 18 yield-related agronomic and physiological traits, along with their drought tolerance indices, in a recombinant inbred line population derived from the XC7 × XC21 cross. Drought stress exerted negative impacts on grain formation and filling, directly leading to significant reductions in grain weight per spike and grain yield per plot, compared with non-stress conditions. Genotyping the recombinant inbred line population using the wheat 90K single nucleotide polymorphism array resulted in the identification of 131 QTLs associated with the 18 traits. Among the identified QTLs, 43 were specifically associated with drought tolerance across the 18 traits, with 6 showing direct linkages to drought tolerance in wheat.

## Figures and Tables

**Figure 1 plants-13-00898-f001:**
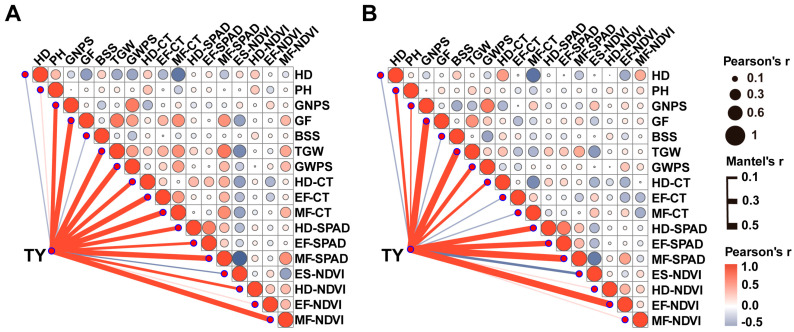
The correlation analysis between total yield and agronomic traits under different treatment conditions. (**A**) The correlation analysis between total yield and agronomic traits with adequate irrigation treatments. (**B**) The correlation analysis between total yield and agronomic traits under drought-stress conditions. TY, total yield; HD, heading date; PH, plant height; GNPS, grain number per spike; GF, grain fullness; BSS, number of basal sterile spikelet; TGW, thousand-grain weight; GWPS, grain weight per spike; HD-CT, canopy temperature measured in heading stage; EF-CT, canopy temperature measured in early filling stage; MF-CT, canopy temperature measured in middle filling stage; HD-SPAD, chlorophyll content measured in heading stage; EF-SPAD, chlorophyll content measured in early filling stage; MF-SPAD, chlorophyll content measured in middle filling stage; ES-NDVI, normalized differential vegetation index measured in elongation stage; HD-NDVI, normalized differential vegetation index measured in heading stage; EF-NDVI, normalized differential vegetation index measured in early filling stage; MF-NDVI, normalized differential vegetation index measured in middle filling stage. Red lines indicate positive correlations, and gray lines indicate negative correlations. The correlations were analyzed by “corrplot” in the R package using the average values presented in Appendix A.

**Figure 2 plants-13-00898-f002:**
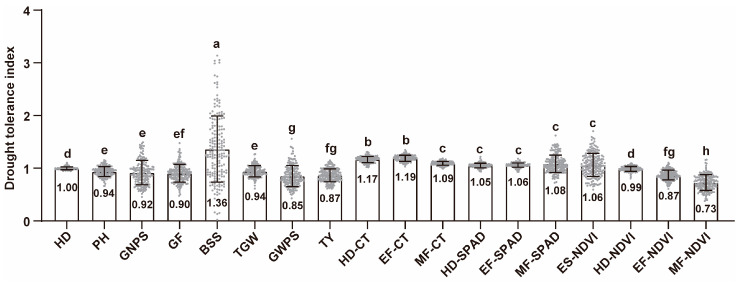
Drought tolerance index(DI) of yield-related agronomic traits and physiological traits. The numbers in the column plot represent the average values. Different letters indicate statistically significant differences among samples, determined using the two-tailed Kruskal–Wallis test with Dunn’s multiple comparison post hoc test (*p* < 0.05).

**Figure 3 plants-13-00898-f003:**
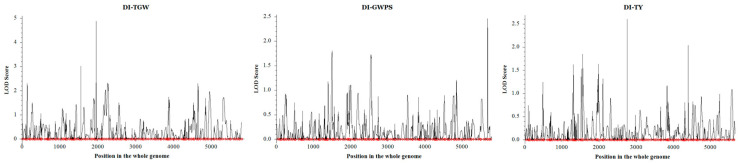
LOD scores of the DIs of TGW, GWPS and TY in the whole genome. DI, drought-resistance index; TGW, thousand-grain weight; GWPS, grain weight per spike; TY, total yield.

**Figure 4 plants-13-00898-f004:**
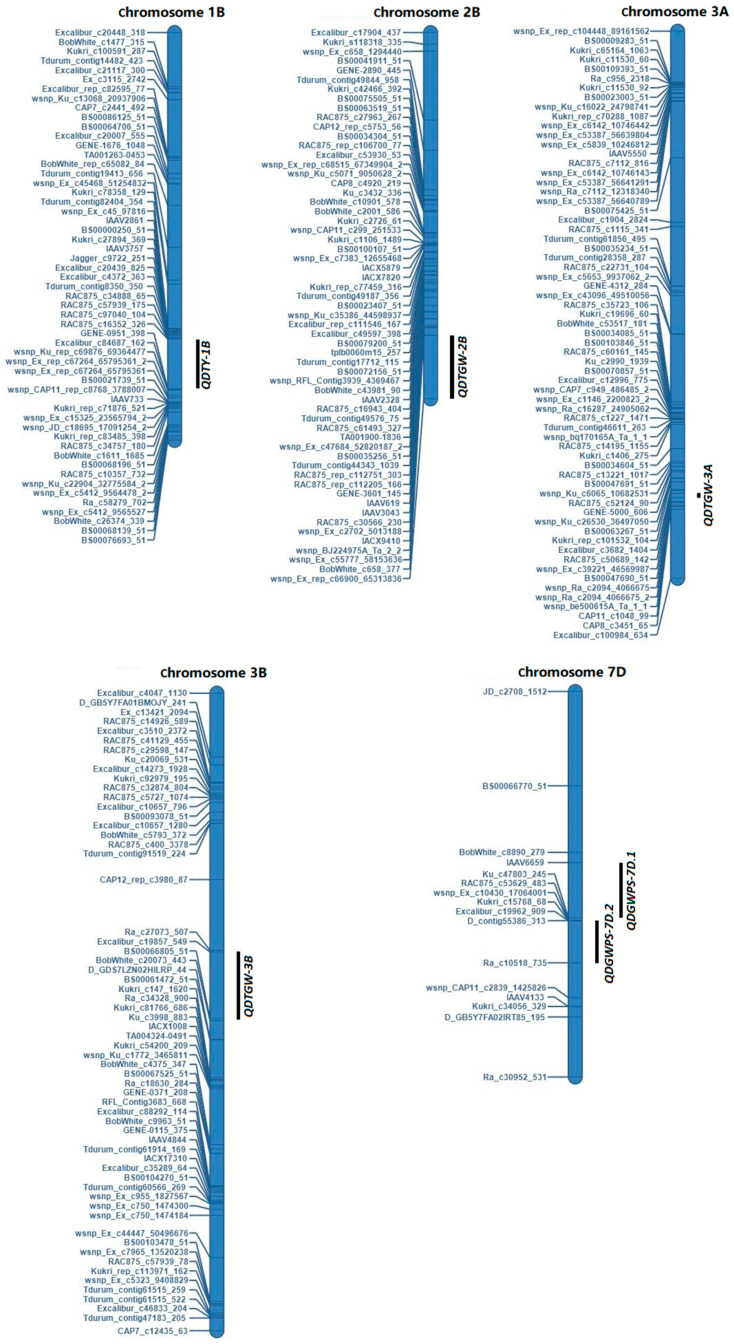
Chromosomal locations of the QTLs that were associated with drought tolerance in wheat.

**Figure 5 plants-13-00898-f005:**
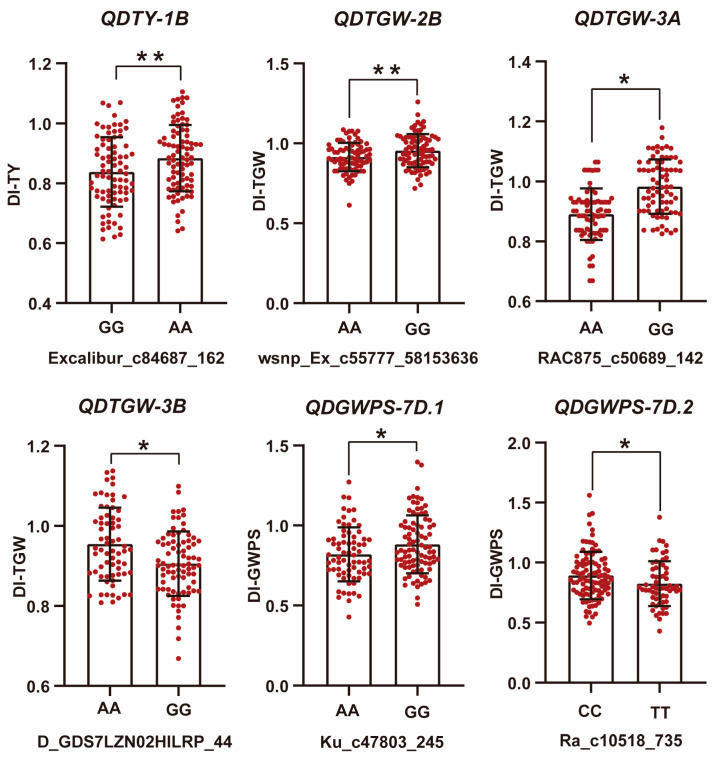
Phenotypic values of the associated SNP markers in the RIL population. * Indicates significant difference at *p* < 0.05, ** Indicates significant difference at *p* < 0.01. DI, drought-resistance index. TGW, thousand-grain weight; GWPS, grain weight per spike; TY, total yield.

**Table 1 plants-13-00898-t001:** The descriptive statistics on agronomic traits of 188 wheat RILs in two treatments.

Traits	Adequate Irrigation				Drought Stress			
	Range	Mean	SD	CV (%)	Range	Mean	SD	CV (%)
HD	38.00–59.00	50.44	4.47	3.79	39.00–59.00	50.28	4.53	3.01
PH	66.90–130.00	102.88	11.41	8.37	63.70–122.00	96.11	10.23	8.48
GNPS	32.30–59.84	44.09	4.98	11.29	28.62–56.30	39.43	4.43	11.22
GF	2.00–8.00	4.78	1.19	12.39	2.00–8.00	4.28	1	11.81
BSS	0.00–4.00	1.41	0.69	34.55	0.00–5.20	1.79	0.86	33.01
TGW	23.20–58.40	40.09	6.38	8.04	20.40–53.60	37.65	5.83	8.06
GWPS	0.38–3.66	1.76	0.5	12.69	0.40–3.32	1.47	0.4	11.11
TY	2.21–12.03	6.75	1.91	8.82	2.29–10.39	5.8	1.53	7.81
HD-CT	13.90–26.90	20.11	1.97	2.55	16.70–31.00	23.42	2.48	2.66
EF-CT	20.85–27.10	23.55	0.91	1.57	22.53–32.57	27.98	1.68	2.29
MF-CT	20.58–32.10	25.06	2.46	1.91	23.20–31.00	27.33	1.66	1.43
HD-SPAD	38.90–69.30	49.26	2.79	3.63	42.40–61.10	51.8	2.45	3.17
EF-SPAD	41.90–60.00	50.82	2.54	3.5	43.40–62.40	53.86	2.52	3.09
MF-SPAD	7.40–63.80	40.16	16.97	4.51	5.10–64.20	43.29	15.57	8.25
ES-NDVI	0.13–0.65	0.31	0.13	9.96	0.11–0.65	0.33	0.14	12.01
HD-NDVI	0.44–0.93	0.85	0.05	2.93	0.63–0.93	0.84	0.04	2.71
EF-NDVI	0.48–0.92	0.79	0.05	3.43	0.38–0.91	0.68	0.11	5.96
MF-NDVI	0.29–0.88	0.68	0.11	7.45	0.18–0.83	0.49	0.13	11.72

SD, standard deviation; CV, coefficient of variation; HD, heading date; PH, plant height; GNPS, grain number per spike; GF, grain fullness; BSS, number of basal sterile spikelet; TGW, thousand-grain weight; GWPS, grain weight per spike; TY, total yield; CT, canopy temperature; SPAD, chlorophyll content; NDVI, normalized differential vegetation index; EF, early filling stage; MF, middle filling stage; ES, elongation stage.

**Table 2 plants-13-00898-t002:** Summary of chromosome assignment, number of SNP marker, map length, marker density and Max Gap of the SNP genetic map.

Chromosome	No. of Markers	Map Distance (cM)	Map Density (marker/cM)	Max Gap (cM)	Gap < 5 cM
1A	88	75.18	1.17	18.44	97.73%
2A	43	134.94	0.32	34.93	90.70%
3A	114	182.13	0.63	44.00	94.74%
4A	85	58.01	1.47	6.88	98.82%
5A	47	107.93	0.44	16.37	89.36%
6A	26	82.94	0.31	28.59	84.62%
7A	59	90.93	0.65	14.48	94.92%
1B	66	170.98	0.39	31.86	90.91%
2B	88	128.31	0.69	34.66	93.18%
3B	88	157.46	0.56	29.13	92.05%
4B	14	148.68	0.09	47.48	64.29%
5B	71	170.78	0.42	35.00	90.14%
6B	34	42.49	0.80	16.02	97.06%
7B	110	109.54	1.00	12.68	96.36%
1D	37	150.87	0.25	39.70	83.78%
2D	19	102.39	0.19	37.67	84.21%
3D	20	73.74	0.27	29.91	80.00%
5D	11	58.68	0.19	58.12	90.91%
6D	15	26.40	0.57	10.54	86.67%
7D	12	115.97	0.10	41.63	58.33%
A genome	462	732.06	0.63	44.00	94.59%
B genome	471	928.24	0.51	47.78	92.36%
D genome	114	528.05	0.22	58.12	81.58%
Total	1047	2188.35	0.48	29.40	92.17%

**Table 3 plants-13-00898-t003:** The environmentally stable QTLs of 18 agronomic traits.

Trait	QTLs	Chr	Left Marker	Right Marker	LOD	PVE (%)	Add	Environment
HD	QHD-2A.1	2A	BS00068196_51	BobWhite_c1611_1685	13.23–13.56	11.77–11.83	0.73	NE1, NE4
	QHD-2A.2	2A	BS00076693_51	BS00068139_51	6.96–9.94	5.53–7.93	−0.55	NE1, NE4
	QHD-2B	2B	wsnp_Ex_c16144_24583060	RAC875_c35399_497	4.23–9.30	7.41–11.32	−0.61	NE1, NE4, D3, D4
	QHD-3D	3D	Excalibur_c27702_282	wsnp_Ra_rep_c116793_96612614	2.71–5.95	0.44–12.6	0.44	NE1, NE2, NE3, NE4, DE1, DE2, DE4
	QHD-5B.1	5B	wsnp_BJ224975A_Ta_2_2	wsnp_Ex_c8543_14357051	5.43–19.87	1.80–34.62	−0.91	NE1, NE2, NE3, NE4, DE2, DE3, DE4
	QHD-5B.2	5B	TA001900-1836	Tdurum_contig49576_75	5.12–5.58	4.36–7.61	−0.54	NE2, NE3, NE4, DE4
	QHD-5B.3	5B	Tdurum_contig44343_1039	wsnp_Ku_c21275_31007309	6.96–7.14	0.73–11.7	−0.55	NE1, DE1, DE2
	QHD-6A	6A	TA004558-1018	tplb0028p11_1104	2.88–4.01	3.04–4.15	0.39	NE3, NE4, DE4
PH	QPH-4B	4B	wsnp_Ex_c26807_36031771	RAC875_rep_c72961_977	3.37–4.74	8.27–9.66	−2.8	NE2, NE3, NE4, DE1, DE2, DE3, DE4
	QPH-5B	5B	wsnp_RFL_Contig3939_4369467	BS00072155_51	2.51–4.47	5.47–10.3	2.5	NE1, NE2, NE4, DE1, DE4
GF	QGF-4B	4B	wsnp_Ex_c26807_36031771	RAC875_rep_c72961_977	3.41–4.95	8.91–10.41	−0.19	NE3, NE4
	QGF-1A	1A	IACX662	BS00012283_51	2.99–4.34	7.73–10.26	0.16	NE4, DE1, DE3, DE4
BSS	QBSS-1A	1A	RAC875_c50864_1921	wsnp_Ex_c57322_59083238	4.69–5.63	9.54–10.45	−0.18	NE3, NE4
	QBSS-5B	5B	wsnp_Ex_c8543_14357051	IAAV619	3.31–4.35	6.40–9.93	−0.16	DE3, NE3, NE4
TGW	QTGW-1A	1A	wsnp_Ex_c33246_41764093_2	wsnp_Ku_c18923_28319203_2	17.14–18.69	8.34–8.38	−2.25	NE2, NE4
	QTGW-2B	2B	Jagger_c1059_300	TA003703-0582	2.54–30.4	4.66–16.2	−2.29	NE2, NE3, NE4
	QTGW-4B	4B	wsnp_Ex_c26807_36031771	RAC875_rep_c72961_977	3.01–5.47	1.75–10.39	−1.08	NE1, NE3, NE4
TY	QTY-1A	1A	BS00022698_51	RAC875_c400_3378	3.15–3.18	7.90–7.96	0.18	NE4, DE1, DE4
HD-SPAD	QHD-SPAD-4B	4B	wsnp_Ex_c26807_36031771	RAC875_rep_c72961_977	3.01–6.00	4.76–14.67	0.61	NE2, NE3, NE4, DE1, DE2, DE3, DE4
	QHD-SPAD-5B	5B	Excalibur_rep_c111129_125	RAC875_c30566_230	4.61–6.11	8.82–8.87	0.67	NE3, NE4
EF-SPAD	QEF-SPAD-5B.1	5B	wsnp_Ra_c13_24911	wsnp_Ex_rep_c68515_67349904_2	2.74–4.72	5.61–8.44	0.50	DE2, DE3, DE4
	QEF-SPAD-5B.2	5B	Excalibur_c6279_381	Kukri_c52049_277	3.25–6.06	6.41–11.2	−0.54	NE4, DE2, DE4
MF-SPAD	QMF-SPAD-3D	3D	RAC875_c35873_1828	Excalibur_c9472_217	3.50–4.28	8.58–8.80	−0.59	NE1, NE4
	QMF-SPAD-5B	5B	Tdurum_contig44343_1039	wsnp_Ku_c21275_31007309	5.77–8.18	9.75–18.92	−2.40	DE3, DE4
ES-NDVI	QES-NDVI-1A.3	1A	Tdurum_contig10036_474	BS00067525_51	6.09–6.59	11.45–14.21	0.13	NE4, DE3, DE4
HD-NDVI	QHD-NDVI-5B	5B	wsnp_Ex_c8543_14357051	IAAV619	3.64–9.44	8.42–19.79	−0.01	NE3, NE4, DE2, DE3, DE4
EF-NDVI	QEF-NDVI-5B.1	5B	wsnp_BJ224975A_Ta_2_2	wsnp_Ex_c8543_14357051	3.28–8.24	7.60–19.72	−0.01	NE3, NE4, DE2, DE4
	QEF-NDVI-5B.2	5B	BS00003944_51	BS00049719_51	2.59–2.65	3.61–6.36	−0.01	DE3, DE4

Chr: Chromosome; PVE: phenotypic variation explained [26]. NE1, NE2, NE3 and NE4 indicate non-stress condition in 2014, 2015, 2016 and the average values, respectively; DE1, DE2, DE3 and DE4 indicate drought stress condition in 2014, 2015, 2016 and the average values, respectively.

## Data Availability

The data presented in this study are available on request from the corresponding author.

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
