# Peer review of "Exploring the Drought Tolerant Quantitative Trait Loci in Spring Wheat"

_plants, 2024, doi:10.3390/plants13060898_

Round 1

Reviewer 1 Report

Comments and Suggestions for Authors

The present paper, Exploring the drought tolerant QTLs in spring wheat, deals with the effect of drought on yield traits in wheat.  Since wheat is one of the most important crops for human nutrition, this research is very useful. The authors compared a total of 18 yield -related agronomic and physiological traits in a total of 186 F8 materials. 

The method used, i.e. QTLs, is very suitable for this type of research with a large amount of important data and information. 

Although the manuscript is very significant, I find several shortcomings that could be corrected. 

1) Please check the formatting of the tables and the quality of all figures so that they are easy to understand for the reader. 

2) The part conclusion is completely missing.

3) Some references are not used appropriately, e.g. citation no 1, not relevant to the content. 

Author Response

Revised in accordance with reviewer report. Please see the attachment.

Reviewer 2 Report

Comments and Suggestions for Authors

Dear authors,

The article 'Exploring the drought tolerant in QTLs in spring wheat' is of interest to plant breeders and genetics, as we are all now under the climate changes pressure. Your research sounds good, but you might want to extend a discussion and whole manuscript in total by adding some new references.

Some minor changes

abstact 

pg 1 line 17 grain yield instead yield

line 37 the same for grain yield

line 38 grain yield components

line 42 grain yield

pg 1- 2 line 46-48 you already mentioned the words plant height, spike length, kernel weight per spike etc. in lines 40-41

So probably it would be suitable to write abbreviations when first mentioned.

pg 2 line 50  SPAD - first time mentioned

pg 2 line  70 why you wrote abbreviations previously when you do not use it (kernels per spike etc)

Results

pg 3 line 80-92 I am not sure is ti suitable to use abbreviations for traits in the whole manuscript, sometimes it is not easy to follow (especially in discussion)

It is difficult to see what is writen at Figure 4

Discussion

Pleas eus emore references to compare your investigation

M and M

it is difficult to foolow up the text with abbreviations. It is ok to use them in the tables.

Author Response

(The authors gave the same response as above.)

Reviewer 3 Report

Comments and Suggestions for Authors

Comments for the manuscript entitled "Exploring the drought tolerant QTLs in spring wheat" submitted by Zhong Wang et al.

The study is focused on unraveling the genetic mechanism that ensures drought tolerance of wheat.

It is known that wheat is a difficult plant to grow in conditions of drought stress, which will increase with global warming. In this regard, the present study focused on the identification of drought-stable quantitative trait loci (QTLs) for wheat yield increase. The research was done on a recombinant inbreb population derived from XinChun7 x XinChun21 cross. This population was subjected to two treatments: non stress and drought-stress conditions. Thus, 131 QTLs associated with 18 agronomic traits and physiological traits were identified.

1027 markers located on all the chromosomes of the three sub-genomes (A, B, D) (except 4D) of the Triticum aestivum genome facilitated the construction of the genetic map necessary to identify the associations between markers and agronomic traits. Thus it was deduced that 43 QTLs (out of a total of 131) were specifically associated with the 18 agronomic and physiological traits. 28 repetitive QTLs were defined as reliable loci.

The drought tolerance index (DI) was also evaluated, showing that grains weight per spike and total yield were the most vulnerable agronomic traits to drought stress. QTLs analysis of the DI concluded that there are 20 QTLs associated with drought tolerance related to agronomic traits related to yield and another 23 QTLs associated with drought tolerance related to physiological traits that condition production yield. Six QTLs were identified as directly related to drought tolerance in wheat.

The polymorphic characterization of the markers used led to the conclusion that certain alleles of there loci (QTLs) could be integrated into wheat cultivars by marker-assisted selection in order to increase drought tolerance and for the stability of the production yield in drought conditions.

The study has practical importance because it deciphers at least partially the genetic control of wheat growth and development, and identifies the traits that support drought tolerance. This information could be used in breeding programs to obtain new wheat cultivars with high drought tolerance and high yield in conditions with and without drought.

My comments are below:

1. In line 38 you should write"ie:" before listing the three components of wheat production yield.

2. During the paper you used the terms: kernel, grains, seeds. In tis regard,  there should be uniformity! Wheat being a cereal has grains or caryopsis. The terms kernel or seed are not appropriate. Attention, because even in the tables and figures you used abbreviation, such as KNS, KWS.

3. Supplementary tables are mentioned in the text, but are not presented.

4. There should be a section "5. Conclusions". You have marked the conclusions throughout the paper, but it would be more appropriate to mention them in a separate section. This would help your readers a lot.

I wish you the best of luck in getting this manuscript published!

Author Response

(The authors gave the same response as above.)

Round 2

Reviewer 1 Report

Comments and Suggestions for Authors

All deficiencies were corrected according to comments.

Author Response

Thanks to the reviewers for their suggestions on the manuscript.